# The Randomized Dependence Coefficient

**David Lopez-Paz, Philipp Hennig, Bernhard Schölkopf**
Max Planck Institute for Intelligent Systems
Spemannstraße 38, Tübingen, Germany
{dlopez,phennig,bs}@tue.mpg.de

## Abstract

We introduce the Randomized Dependence Coefficient (RDC), a measure of non-linear dependence between random variables of arbitrary dimension based on the Hirschfeld-Gebelein-Rényi Maximum Correlation Coefficient. RDC is defined in terms of correlation of random non-linear copula projections; it is invariant with respect to marginal distribution transformations, has low computational cost and is easy to implement: just five lines of R code, included at the end of the paper.

## 1  Introduction

Measuring statistical dependence between random variables is a fundamental problem in statistics. Commonly used measures of dependence, Pearson's rho, Spearman's rank or Kendall's tau are computationally efficient and theoretically well understood, but consider only a limited class of association patterns, like linear or monotonically increasing functions. The development of non-linear dependence measures is challenging because of the radically larger amount of possible association patterns.

Despite these difficulties, many non-linear statistical dependence measures have been developed recently. Examples include the Alternating Conditional Expectations or *backfitting algorithm* (ACE) [2, 9], Kernel Canonical Correlation Analysis (KCCA) [1], (Copula) Hilbert-Schmidt Independence Criterion (CHSIC, HSIC) [6, 5, 15], Distance or Brownian Correlation (dCor) [24, 23] and the Maximal Information Coefficient (MIC) [18]. However, these methods exhibit high computational demands (at least quadratic costs in the number of samples for KCCA, HSIC, CHSIC, dCor or MIC), are limited to measuring dependencies between scalar random variables (ACE, MIC) or can be difficult to implement (ACE, MIC).

This paper develops the *Randomized Dependence Coefficient* (RDC), an estimator of the Hirschfeld-Gebelein-Rényi Maximum Correlation Coefficient (HGR) addressing the issues listed above. RDC defines dependence between two random variables as the largest canonical correlation between random non-linear projections of their respective empirical copula-transformations. RDC is invariant to monotonically increasing transformations, operates on random variables of arbitrary dimension, and has computational cost of $O(n \log n)$ with respect to the sample size. Moreover, it is easy to implement: just five lines of R code, included in Appendix A.

The following Section reviews the classic work of Alfréd Rényi [17], who proposed seven desirable fundamental properties of dependence measures, proved to be satisfied by the Hirschfeld-Gebelein-Rényi's Maximum Correlation Coefficient (HGR). Section 3 introduces the Randomized Dependence Coefficient as an estimator designed in the spirit of HGR, since HGR itself is computationally intractable. Properties of RDC and its relationship to other non-linear dependence measures are analysed in Section 4. Section 5 validates the empirical performance of RDC on a series of numerical experiments on both synthetic and real-world data.

## 2 Hirschfeld-Gebelein-Rényi's Maximum Correlation Coefficient

In 1959 [17], Alfréd Rényi argued that a measure of dependence $\rho^* : \mathcal{X} \times \mathcal{Y} \to [0,1]$ between random variables $X \in \mathcal{X}$ and $Y \in \mathcal{Y}$ should satisfy seven fundamental properties:

1. $\rho^*(X, Y)$ is defined for any pair of non-constant random variables $X$ and $Y$.
2. $\rho^*(X, Y) = \rho^*(Y, X)$
3. $0 \leq \rho^*(X, Y) \leq 1$
4. $\rho^*(X, Y) = 0$ iff $X$ and $Y$ are statistically independent.
5. For bijective Borel-measurable functions $f, g : \mathbb{R} \to \mathbb{R}$, $\rho^*(X, Y) = \rho^*(f(X), g(Y))$.
6. $\rho^*(X, Y) = 1$ if for Borel-measurable functions $f$ or $g$, $Y = f(X)$ or $X = g(Y)$.
7. If $(X, Y) \sim \mathcal{N}(\boldsymbol{\mu}, \boldsymbol{\Sigma})$, then $\rho^*(X, Y) = |\rho(X, Y)|$, where $\rho$ is the correlation coefficient.

Rényi also showed the *Hirschfeld-Gebelein-Rényi Maximum Correlation Coefficient* (HGR) [3, 17] to satisfy all these properties. HGR was defined by Gebelein in 1941 [3] as the supremum of Pearson's correlation coefficient $\rho$ over all Borel-measurable functions $f, g$ of finite variance:

$$\text{hgr}(X, Y) = \sup_{f,g} \rho(f(X), g(Y)), \tag{1}$$

Since the supremum in (1) is over an infinite-dimensional space, HGR is not computable. It is an abstract concept, not a practical dependence measure. In the following we propose a scalable estimator with the same structure as HGR: the Randomized Dependence Coefficient.

## 3 Randomized Dependence Coefficient

The *Randomized Dependence Coefficient* (RDC) measures the dependence between random samples $\boldsymbol{X} \in \mathbb{R}^{p \times n}$ and $\boldsymbol{Y} \in \mathbb{R}^{q \times n}$ as the largest canonical correlation between $k$ randomly chosen non-linear projections of their copula transformations. Before Section 3.4 defines this concept formally, we describe the three necessary steps to construct the RDC statistic: copula-transformation of each of the two random samples (Section 3.1), projection of the copulas through $k$ randomly chosen non-linear maps (Section 3.2) and computation of the largest canonical correlation between the two sets of non-linear random projections (Section 3.3). Figure 1 offers a sketch of this process.

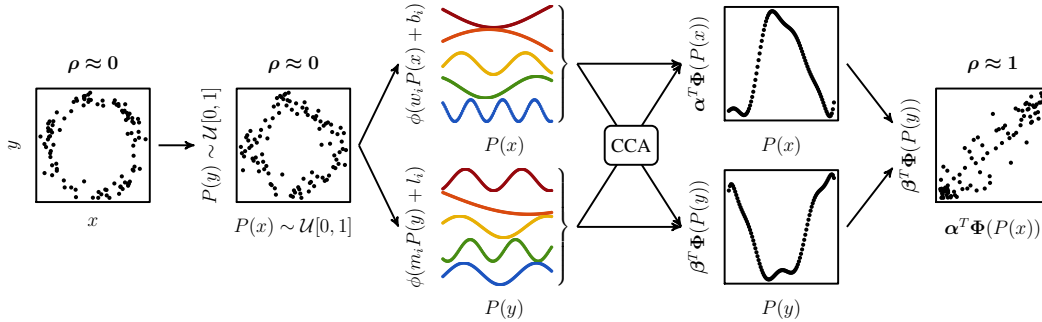

Figure 1: RDC computation for a simple set of samples $\{(x_i, y_i)\}_{i=1}^{100}$ drawn from a noisy circular pattern: The samples are used to estimate the copula, then mapped with randomly drawn non-linear functions. The RDC is the largest canonical correlation between these non-linear projections.

### 3.1 Estimation of Copula-Transformations

To achieve invariance with respect to transformations on marginal distributions (such as shifts or rescalings), we operate on the *empirical copula transformation* of the data [14, 15]. Consider a random vector $\boldsymbol{X} = (X_1, \ldots, X_d)$ with continuous marginal cumulative distribution functions (cdfs) $P_i$, $1 \leq i \leq d$. Then the vector $\boldsymbol{U} = (U_1, \ldots, U_d) := \boldsymbol{P}(\boldsymbol{X}) = (P_1(X_1), \ldots, P_d(X_d))$, known as the *copula transformation*, has uniform marginals:

**Theorem 1.** *(Probability Integral Transform [14]) For a random variable $X$ with cdf $P$, the random variable $U := P(X)$ is uniformly distributed on $[0, 1]$.*

The random variables $U_1, \ldots, U_d$ are known as the observation ranks of $X_1, \ldots, X_d$. Crucially, $\boldsymbol{U}$ preserves the dependence structure of the original random vector $\boldsymbol{X}$, but ignores each of its $d$ marginal forms [14]. The joint distribution of $\boldsymbol{U}$ is known as the copula of $\boldsymbol{X}$:

**Theorem 2.** *(Sklar [20]) Let the random vector $\boldsymbol{X} = (X_1, \ldots, X_d)$ have continuous marginal cdfs $P_i$, $1 \leq i \leq d$. Then, the joint cumulative distribution of $\boldsymbol{X}$ is uniquely expressed as:*

$$P(X_1, \ldots, X_d) = C(P_1(X_1), \ldots, P_d(X_d)), \tag{2}$$

*where the distribution $C$ is known as the copula of $\boldsymbol{X}$.*

A practical estimator of the univariate cdfs $P_1, \ldots, P_d$ is the *empirical cdf*:

$$P_n(x) := \frac{1}{n} \sum_{i=1}^{n} \mathbb{I}(X_i \leq x), \tag{3}$$

which gives rise to the *empirical copula transformations* of a multivariate sample:

$$\boldsymbol{P}_n(\boldsymbol{x}) = [P_{n,1}(x_1), \ldots, P_{n,d}(x_d)]. \tag{4}$$

The Massart-Dvoretzky-Kiefer-Wolfowitz inequality [13] can be used to show that empirical copula transformations converge fast to the true transformation as the sample size increases:

**Theorem 3.** *(Convergence of the empirical copula, [15, Lemma 7]) Let $\boldsymbol{X}_1, \ldots, \boldsymbol{X}_n$ be an i.i.d. sample from a probability distribution over $\mathbb{R}^d$ with marginal cdf's $P_1, \ldots, P_d$. Let $\boldsymbol{P}(\boldsymbol{X})$ be the copula transformation and $\boldsymbol{P}_n(\boldsymbol{X})$ the empirical copula transformation. Then, for any $\epsilon > 0$:*

$$\Pr\left[ \sup_{\boldsymbol{x} \in \mathbb{R}^d} \| \boldsymbol{P}(\boldsymbol{x}) - \boldsymbol{P}_n(\boldsymbol{x}) \|_2 > \epsilon \right] \leq 2d \exp\left( -\frac{2n\epsilon^2}{d} \right). \tag{5}$$

Computing $\boldsymbol{P}_n(\boldsymbol{X})$ involves sorting the marginals of $\boldsymbol{X} \in \mathbb{R}^{d \times n}$, thus $O(dn \log(n))$ operations.

## 3.2  Generation of Random Non-Linear Projections

The second step of the RDC computation is to augment the empirical copula transformations with non-linear projections, so that linear methods can subsequently be used to capture non-linear dependencies on the original data. This is a classic idea also used in other areas, particularly in regression. In an elegant result, Rahimi and Recht [16] proved that linear regression on random, non-linear projections of the original feature space can generate high-performance regressors:

**Theorem 4.** *(Rahimi-Recht) Let $p$ be a distribution on $\Omega$ and $|\phi(\boldsymbol{x}; \boldsymbol{w})| \leq 1$. Let $\mathcal{F} = \left\{ f(\boldsymbol{x}) = \int_\Omega \alpha(\boldsymbol{w}) \phi(\boldsymbol{x}; \boldsymbol{w}) \mathrm{d}\boldsymbol{w} \, \big| \, |\alpha(\boldsymbol{w})| \leq C p(\boldsymbol{w}) \right\}$. Draw $\boldsymbol{w}_1, \ldots, \boldsymbol{w}_k$ iid from $p$. Further let $\delta > 0$, and $c$ be some L-Lipschitz loss function, and consider data $\{\boldsymbol{x}_i, y_i\}_{i=1}^{n}$ drawn iid from some arbitrary $P(\boldsymbol{X}, Y)$. The $\alpha_1, \ldots, \alpha_k$ for which $f_k(\boldsymbol{x}) = \sum_{i=1}^{k} \alpha_i \phi(\boldsymbol{x}; \boldsymbol{w}_i)$ minimizes the empirical risk $c(f_k(\boldsymbol{x}), y)$ has a distance from the c-optimal estimator in $\mathcal{F}$ bounded by*

$$\mathbb{E}_P[c(f_k(\boldsymbol{x}), y)] - \min_{f \in \mathcal{F}} \mathbb{E}_P[c(f(\boldsymbol{x}), y)] \leq O\left( \left( \frac{1}{\sqrt{n}} + \frac{1}{\sqrt{k}} \right) LC \sqrt{\log \frac{1}{\delta}} \right) \tag{6}$$

*with probability at least $1 - 2\delta$.*

Intuitively, Theorem 4 states that randomly selecting $\boldsymbol{w}_i$ in $\sum_{i=1}^{k} \alpha_i \phi(\boldsymbol{x}; \boldsymbol{w}_i)$ instead of optimising them causes only bounded error.

The choice of the non-linearities $\phi : \mathbb{R} \rightarrow \mathbb{R}$ is the main and unavoidable assumption in RDC. This choice is a well-known problem common to all non-linear regression methods and has been studied extensively in the theory of regression as the selection of reproducing kernel Hilbert space [19, §3.13]. The only way to favour one such family and distribution over another is to use prior assumptions about which kind of distributions the method will typically have to analyse.

We use random features instead of the Nyström method because of their smaller memory and computation requirements [11]. In our experiments, we will use sinusoidal projections, $\phi(\boldsymbol{w}^T\boldsymbol{x} + b) := \sin(\boldsymbol{w}^T\boldsymbol{x} + b)$. Arguments favouring this choice are that shift-invariant kernels are approximated with these features when using the appropriate random parameter sampling distribution [16], [4, p. 208] [22, p. 24], and that functions with absolutely integrable Fourier transforms are approximated with $L_2$ error below $O(1/\sqrt{k})$ by $k$ of these features [10].

Let the random parameters $\boldsymbol{w}_i \sim \mathcal{N}(\boldsymbol{0}, s\boldsymbol{I})$, $b_i \sim \mathcal{N}(0, s)$. Choosing $\boldsymbol{w}_i$ to be Normal is analogous to the use of the Gaussian kernel for HSIC, CHSIC or KCCA [16]. Tuning $s$ is analogous to selecting the kernel width, that is, to regularize the non-linearity of the random projections.

Given a data collection $\boldsymbol{X} = (\boldsymbol{x}_1, \ldots, \boldsymbol{x}_n)$, we will denote by

$$\boldsymbol{\Phi}(\boldsymbol{X}; k, s) := \left( \begin{array}{ccc} \phi(\boldsymbol{w}_1^T\boldsymbol{x}_1 + b_1) & \cdots & \phi(\boldsymbol{w}_k^T\boldsymbol{x}_1 + b_k) \\ \vdots & \vdots & \vdots \\ \phi(\boldsymbol{w}_1^T\boldsymbol{x}_n + b_1) & \cdots & \phi(\boldsymbol{w}_k^T\boldsymbol{x}_n + b_k) \end{array} \right)^T \tag{7}$$

the $k-$th order random non-linear projection from $\boldsymbol{X} \in \mathbb{R}^{d \times n}$ to $\boldsymbol{\Phi}(\boldsymbol{X}; k, s) \in \mathbb{R}^{k \times n}$. The computational complexity of computing $\boldsymbol{\Phi}(\boldsymbol{X}; k, s)$ with naive matrix multiplications is $O(kdn)$. However, recent techniques using fast Walsh-Hadamard transforms [11] allow computing these feature expansions within a computational cost of $O(k\log(d)n)$ and $O(k)$ storage.

### 3.3 Computation of Canonical Correlations

The final step of RDC is to compute the linear combinations of the augmented empirical copula transformations that have maximal correlation. Canonical Correlation Analysis (CCA, [7]) is the calculation of pairs of basis vectors $(\boldsymbol{\alpha}, \boldsymbol{\beta})$ such that the projections $\boldsymbol{\alpha}^T\boldsymbol{X}$ and $\boldsymbol{\beta}^T\boldsymbol{Y}$ of two random samples $\boldsymbol{X} \in \mathbb{R}^{p \times n}$ and $\boldsymbol{Y} \in \mathbb{R}^{q \times n}$ are maximally correlated. The correlations between the projected (or canonical) random samples are referred to as canonical correlations. There exist up to $\max(\text{rank}(\boldsymbol{X}), \text{rank}(\boldsymbol{Y}))$ of them. Canonical correlations $\rho^2$ are the solutions to the eigenproblem:

$$\left( \begin{array}{cc} \boldsymbol{0} & \boldsymbol{C}_{xx}^{-1}\boldsymbol{C}_{xy} \\ \boldsymbol{C}_{yy}^{-1}\boldsymbol{C}_{yx} & \boldsymbol{0} \end{array} \right) \left( \begin{array}{c} \boldsymbol{\alpha} \\ \boldsymbol{\beta} \end{array} \right) = \rho^2 \left( \begin{array}{c} \boldsymbol{\alpha} \\ \boldsymbol{\beta} \end{array} \right), \tag{8}$$

where $\boldsymbol{C}_{xy} = \text{cov}(\boldsymbol{X}, \boldsymbol{Y})$ and the matrices $\boldsymbol{C}_{xx}$ and $\boldsymbol{C}_{yy}$ are assumed to be invertible. Therefore, the largest canonical correlation $\rho_1$ between $\boldsymbol{X}$ and $\boldsymbol{Y}$ is the supremum of the correlation coefficients over their linear projections, that is: $\rho_1(\boldsymbol{X}, \boldsymbol{Y}) = \sup_{\boldsymbol{\alpha}, \boldsymbol{\beta}} \rho(\boldsymbol{\alpha}^T\boldsymbol{X}, \boldsymbol{\beta}^T\boldsymbol{Y})$.

When $p, q \ll n$, the cost of CCA is dominated by the estimation of the matrices $\boldsymbol{C}_{xx}$, $\boldsymbol{C}_{yy}$ and $\boldsymbol{C}_{xy}$, hence being $O((p+q)^2n)$ for two random variables of dimensions $p$ and $q$, respectively.

### 3.4 Formal Definition or RDC

Given the random samples $\boldsymbol{X} \in \mathbb{R}^{p \times n}$ and $\boldsymbol{Y} \in \mathbb{R}^{q \times n}$ and the parameters $k \in \mathbb{N}_+$ and $s \in \mathbb{R}_+$, the Randomized Dependence Coefficient between $\boldsymbol{X}$ and $\boldsymbol{Y}$ is defined as:

$$\text{rdc}(\boldsymbol{X}, \boldsymbol{Y}; k, s) := \sup_{\boldsymbol{\alpha}, \boldsymbol{\beta}} \rho\left(\boldsymbol{\alpha}^T\boldsymbol{\Phi}(\boldsymbol{P}(\boldsymbol{X}); k, s), \boldsymbol{\beta}^T\boldsymbol{\Phi}(\boldsymbol{P}(\boldsymbol{Y}); k, s)\right). \tag{9}$$

## 4 Properties of RDC

**Computational complexity:** In the typical setup (very large $n$, large $p$ and $q$, small $k$) the computational complexity of RDC is dominated by the calculation of the copula-transformations. Hence, we achieve a cost in terms of the sample size of $O((p+q)n\log n + kn\log(pq) + k^2n) \approx O(n\log n)$.

**Ease of implementation:** An implementation of RDC in R is included in the Appendix A.

**Relationship to the HGR coefficient:** It is tempting to wonder whether RDC is a consistent, or even an efficient estimator of the HGR coefficient. However, a simple experiment shows that it is not desirable to approximate HGR exactly on finite datasets: Consider $p(X, Y) = \mathcal{N}(x; 0, 1)\mathcal{N}(y; 0, 1)$

which is independent, thus, by both Rényi's 4th and 7th properties, has $\mathrm{hgr}(X,Y) = 0$. However, for finitely many $N$ samples from $p(X,Y)$, almost surely, values in both $X$ and $Y$ are pairwise different and separated by a finite difference. So there exist continuous (thus Borel measurable) functions $f(X)$ and $g(Y)$ mapping both $X$ and $Y$ to the sorting ranks of $Y$, i.e. $f(x_i) = g(y_i) \; \forall (x_i, y_i) \in (\boldsymbol{X}, \boldsymbol{Y})$. Therefore, the finite-sample version of Equation (1) is constant and equal to "1" for continuous random variables. Meaningful measures of dependence from finite samples thus must rely on some form of regularization. RDC achieves this by approximating the space of Borel measurable functions with the restricted function class $\mathcal{F}$ from Theorem 4:

Assume the optimal transformations $f$ and $g$ (Equation 1) to belong to the Reproducing Kernel Hilbert Space $\mathcal{F}$ (Theorem 4), with associated shift-invariant, positive semi-definite kernel function $k(\boldsymbol{x}, \boldsymbol{x}') = \langle \boldsymbol{\phi}(\boldsymbol{x}), \boldsymbol{\phi}(\boldsymbol{x}') \rangle_{\mathcal{F}} \leq 1$. Then, with probability greater than $1 - 2\delta$:

$$\mathrm{hgr}(\boldsymbol{X}, \boldsymbol{Y}; \mathcal{F}) - \mathrm{rdc}(\boldsymbol{X}, \boldsymbol{Y}; k) = O\left( \left( \frac{\|\boldsymbol{m}\|_F}{\sqrt{n}} + \frac{LC}{\sqrt{k}} \right) \sqrt{\log \frac{1}{\delta}} \right), \qquad (10)$$

where $\boldsymbol{m} := \boldsymbol{\alpha}\boldsymbol{\alpha}^T + \boldsymbol{\beta}\boldsymbol{\beta}^T$ and $n$, $k$ denote the sample size and number of random features. The bound (10) is the sum of two errors. The error $O(1/\sqrt{n})$ is due to the convergence of CCA's largest eigenvalue in the finite sample size regime. This result [8, Theorem 6] is originally obtained by posing CCA as a least squares regression on the product space induced by the feature map $\boldsymbol{\psi}(\boldsymbol{x}, \boldsymbol{y}) = [\boldsymbol{\phi}(\boldsymbol{x})\boldsymbol{\phi}(\boldsymbol{x})^T, \boldsymbol{\phi}(\boldsymbol{y})\boldsymbol{\phi}(\boldsymbol{y})^T, \sqrt{2}\boldsymbol{\phi}(\boldsymbol{x})\boldsymbol{\phi}(\boldsymbol{y})^T]^T$. Because of approximating $\boldsymbol{\psi}$ with $k$ random features, an additional error $O(1/\sqrt{k})$ is introduced in the least squares regression [16, Lemma 3]. Therefore, an equivalence between RDC and KCCA is established if RDC uses an infinite number of sinusoidal features, the random sampling distribution is set to the inverse Fourier transform of the shift-invariant kernel used by KCCA and the copula-transformations are discarded. However, when $k \geq n$ regularization is needed to avoid spurious perfect correlations, as discussed above.

**Relationship to other estimators:** Table 1 summarizes several state-of-the-art dependence measures showing, for each measure, whether it allows for general non-linear dependence estimation, handles multidimensional random variables, is invariant with respect to changes in marginal distributions, returns a statistic in $[0, 1]$, satisfy Rényi's properties (Section 2), and how many parameters it requires. As parameters, we here count the kernel function for kernel methods, the basis function and number of random features for RDC, the stopping tolerance for ACE and the search-grid size for MIC, respectively. Finally, the table lists computational complexities with respect to sample size.

When using random features $\phi$ linear for some neighbourhood around zero (like sinusoids or sigmoids), RDC converges to Spearman's rank correlation coefficient as $s \to 0$, for any $k$.

Table 1: Comparison between non-linear dependence measures.

| Name of Coeff. | Non-Linear | Vector Inputs | Marginal Invariant | Renyi's Properties | Coeff. $\in [0,1]$ | # Par. | Comp. Cost |
|---|---|---|---|---|---|---|---|
| Pearson's $\rho$ | × | × | × | × | ✓ | 0 | $n$ |
| Spearman's $\rho$ | × | × | ✓ | × | ✓ | 0 | $n \log n$ |
| Kendall's $\tau$ | × | × | ✓ | × | ✓ | 0 | $n \log n$ |
| CCA | × | ✓ | × | × | ✓ | 0 | $n$ |
| KCCA [1] | ✓ | ✓ | × | × | ✓ | 1 | $n^3$ |
| ACE [2] | ✓ | × | × | ✓ | ✓ | 1 | $n$ |
| MIC [18] | ✓ | × | × | × | ✓ | 1 | $n^{1.2}$ |
| dCor [24] | ✓ | ✓ | × | × | ✓ | 1 | $n^2$ |
| HSIC [5] | ✓ | ✓ | × | × | × | 1 | $n^2$ |
| CHSIC [15] | ✓ | ✓ | ✓ | × | × | 1 | $n^2$ |
| **RDC** | ✓ | ✓ | ✓ | ✓ | ✓ | 2 | $n \log n$ |

**Testing for independence with RDC:** Consider the hypothesis "the two sets of non-linear projections are mutually uncorrelated". Under normality assumptions (or large sample sizes), Bartlett's approximation [12] can be used to show $\left( \frac{2k+3}{2} - n \right) \log \prod_{i=1}^{k} (1 - \rho_i^2) \sim \chi_{k^2}^2$, where $\rho_1, \dots, \rho_k$ are the

canonical correlations between $\mathbf{\Phi}(\boldsymbol{P}(\boldsymbol{X}); k, s)$ and $\mathbf{\Phi}(\boldsymbol{P}(\boldsymbol{Y}); k, s)$. Alternatively, non-parametric asymptotic distributions can be obtained from the spectrum of the inner products of the non-linear random projection matrices [25, Theorem 3].

# 5  Experimental Results

We performed experiments on both synthetic and real-world data to validate the empirical performance of RDC versus the non-linear dependence measures listed in Table 1. In some experiments we do not compare against to KCCA because we were unable to find a good set of hyperparameters.

**Parameter selection:**  For RDC, the number of random features is set to $k = 20$ for both random samples, since no significant improvements were observed for larger values. The random feature sampling parameter $s$ is more crucial, and set as follows: when the marginals of $\boldsymbol{u}$ are standard uniforms, $\boldsymbol{w} \sim \mathcal{N}(\mathbf{0}, s\boldsymbol{I})$ and $b \sim \mathcal{N}(0, s)$, then $\mathbb{V}[\boldsymbol{w}^T\boldsymbol{u} + b] = s\left(1 + \frac{d}{3}\right)$; therefore, we opt to set $s$ to a linear scaling of the input variable dimensionality. In all our experiments $s = \frac{1}{6d}$ worked well. The development of better methods to set the parameters of RDC is left as future work.

HSIC and CHSIC use Gaussian kernels $k(\boldsymbol{z}, \boldsymbol{z}') = exp(-\gamma\|\boldsymbol{z} - \boldsymbol{z}'\|_2^2)$ with $\gamma^{-1}$ set to the euclidean distance median of each sample [5]. MIC's search-grid size is set to $B(n) = n^{0.6}$ as recommended by the authors [18], although speed improvements are achieved when using lower values. ACE's tolerance is set to $\epsilon = 0.01$, default value in the R package `acepack`.

## 5.1  Synthetic Data

**Resistance to additive noise:**  We define the *power* of a dependence measure as its ability to discern between dependent and independent samples that share equal marginal forms. In the spirit of Simon and Tibshirani[1], we conducted experiments to estimate the power of RDC as a measure of non-linear dependence. We chose 8 bivariate association patterns, depicted inside little boxes in Figure 3. For each of the 8 association patterns, 500 repetitions of 500 samples were generated, in which the input sample was uniformly distributed on the unit interval. Next, we regenerated the input sample randomly, to generate independent versions of each sample with equal marginals. Figure 3 shows the power for the discussed non-linear dependence measures as the variance of some zero-mean Gaussian additive noise increases from $1/30$ to $3$. RDC shows worse performance in the linear association pattern due to overfitting and in the step-function due to the smoothness prior induced by the sinusoidal features, but has good performance in non-functional association patterns.

**Running times:**  Table 2 shows running times for the considered non-linear dependence measures on scalar, uniformly distributed, independent samples of sizes $\{10^3, \ldots, 10^6\}$ when averaging over 100 runs. Single runs above ten minutes were cancelled. Pearson's $\rho$, ACE, dCor, KCCA and MIC are implemented in C, while RDC, HSIC and CHSIC are implemented as interpreted R code. KCCA is approximated using incomplete Cholesky decompositions as described in [1].

Table 2: Average running times (in seconds) for dependence measures on versus sample sizes.

| sample size | Pearson's $\rho$ | RDC | ACE | KCCA | dCor | HSIC | CHSIC | MIC |
|---|---|---|---|---|---|---|---|---|
| 1,000 | 0.0001 | 0.0047 | 0.0080 | 0.402 | 0.3417 | 0.3103 | 0.3501 | 1.0983 |
| 10,000 | 0.0002 | 0.0557 | 0.0782 | 3.247 | 59.587 | 27.630 | 29.522 | — |
| 100,000 | 0.0071 | 0.3991 | 0.5101 | 43.801 | — | — | — | — |
| 1,000,000 | 0.0914 | 4.6253 | 5.3830 | — | — | — | — | — |

**Value of statistic in** $[0, 1]$**:**  Figure 4 shows RDC, ACE, dCor, MIC, Pearson's $\rho$, Spearman's rank and Kendall's $\tau$ dependence estimates for 14 different associations of two scalar random samples. RDC scores values close to one on all the proposed dependent associations, whilst scoring values close to zero for the independent association, depicted last. When the associations are Gaussian (first row), RDC scores values close to the Pearson's correlation coefficient (Section 2, 7th property).

## 5.2 Feature Selection in Real-World Data

We performed greedy feature selection via dependence maximization [21] on eight real-world datasets. More specifically, we attempted to construct the subset of features $\mathcal{G} \subset \mathcal{X}$ that minimizes the normalized mean squared regression error (NMSE) of a Gaussian process regressor. We do so by selecting the feature $x^{(i)}$ maximizing dependence between the feature set $\mathcal{G}_i = \{\mathcal{G}_{i-1}, x^{(i)}\}$ and the target variable $y$ at each iteration $i \in \{1, \dots 10\}$, such that $\mathcal{G}_0 = \{\emptyset\}$ and $x^{(i)} \notin \mathcal{G}_{i-1}$.

We considered 12 heterogeneous datasets, obtained from the UCI dataset repository[2], the Gaussian process web site Data[3] and the Machine Learning data set repository[4]. Random training/test partitions are computed to be disjoint and equal sized.

Since $\mathcal{G}$ can be multi-dimensional, we compare RDC to the non-linear methods dCor, HSIC and CHSIC. Given their quadratic computational demands, dCor, HSIC and CHSIC use up to $1,000$ points when measuring dependence; this constraint only applied on the `sarcos` and `abalone` datasets. Results are averages of 20 random training/test partitions.

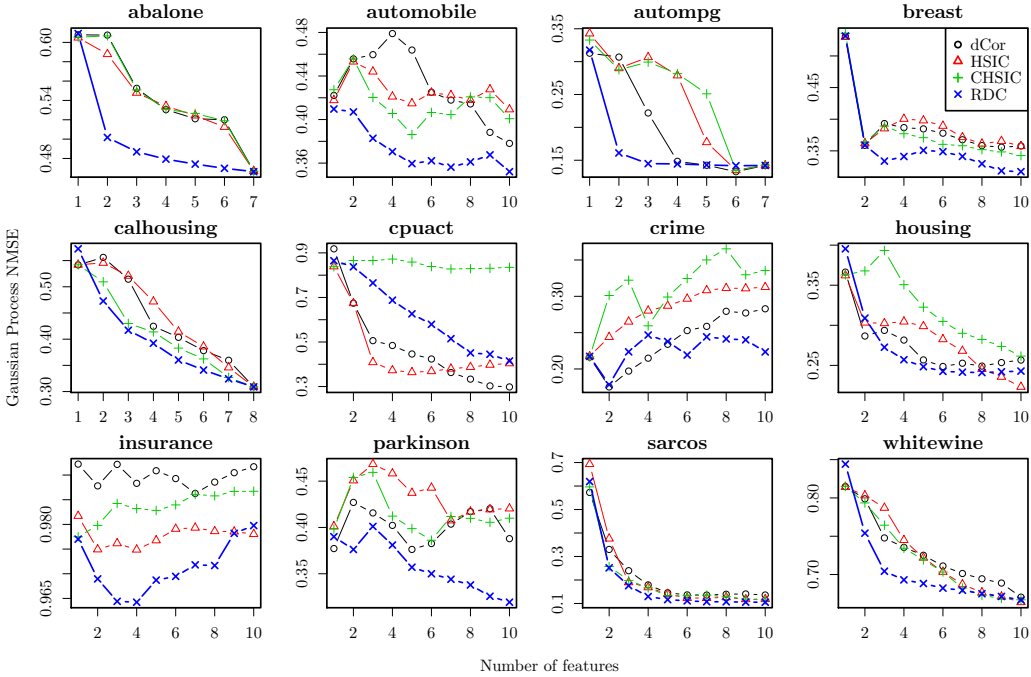

Figure 2: Feature selection experiments on real-world datasets.

Figure 2 summarizes the results for all datasets and algorithms as the number of selected features increases. RDC performs best in most datasets, with much lower running time than its contenders.

## 6 Conclusion

We have presented the randomized dependence coefficient, a lightweight non-linear measure of dependence between multivariate random samples. Constructed as a finite-dimensional estimator in the spirit of the Hirschfeld-Gebelein-Rényi maximum correlation coefficient, RDC performs well empirically, is scalable to very large datasets, and is easy to adapt to concrete problems.

We thank fruitful discussions with Alberto Suárez, Theofanis Karaletsos and David Reshef.

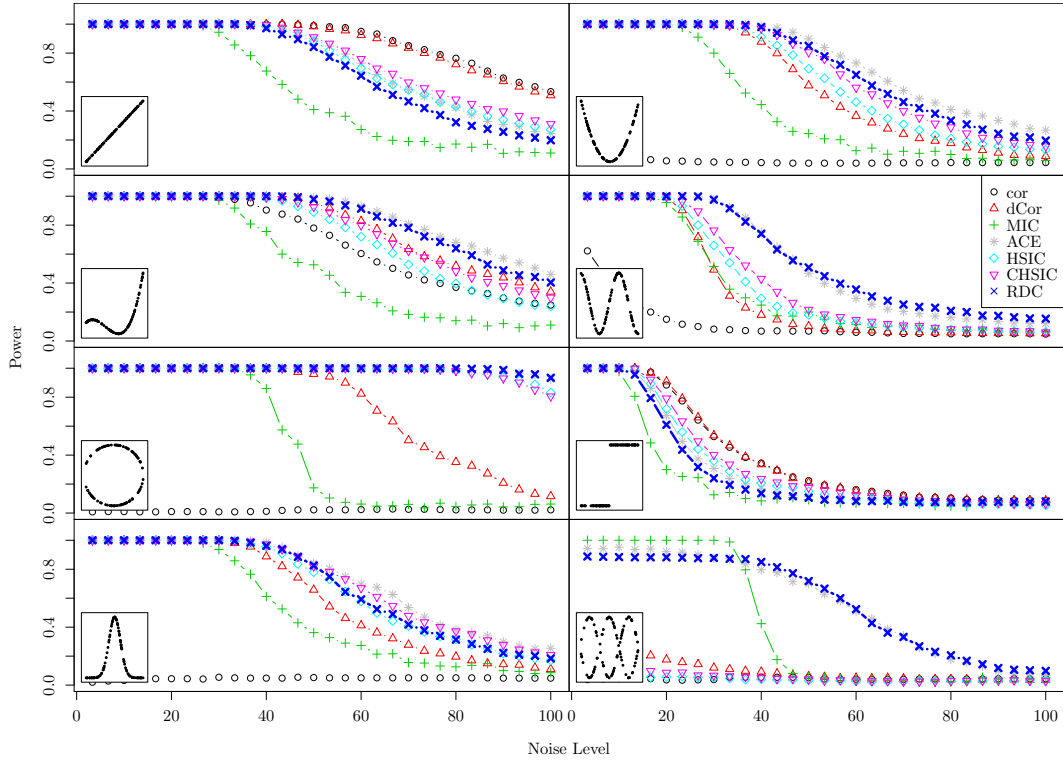

Figure 3: Power of discussed measures on several bivariate association patterns as noise increases. Insets show the noise-free form of each association pattern.

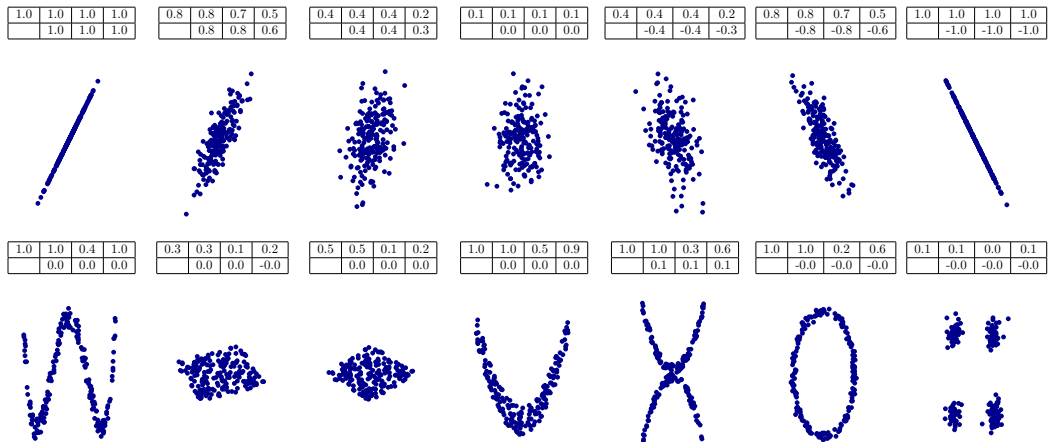

Figure 4: RDC, ACE, dCor, MIC, Pearson's $\rho$, Spearman's rank and Kendall's $\tau$ estimates (numbers in tables above plots, in that order) for several bivariate association patterns.

## A   R Source Code

```
rdc <- function(x,y,k=20,s=1/6,f=sin) {
  x <- cbind(apply(as.matrix(x),2,function(u)rank(u)/length(u)),1)
  y <- cbind(apply(as.matrix(y),2,function(u)rank(u)/length(u)),1)
  x <- s/ncol(x)*x%*%matrix(rnorm(ncol(x)*k),ncol(x))
  y <- s/ncol(y)*y%*%matrix(rnorm(ncol(y)*k),ncol(y))
  cancor(cbind(f(x),1),cbind(f(y),1))$cor[1]
}
```

## Footnotes

[1] http://www-stat.stanford.edu/~tibs/reshef/comment.pdf

[2]`http://www.ics.uci.edu/~mlearn`

[3]`http://www.gaussianprocess.org/gpml/data/`

[4]`http://www.mldata.org`

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
