[Reviews · NeurIPS 2013]

Submitted by Assigned_Reviewer_4

The paper introduces a new method called RDC to measure the statistical dependence between random variables. It combines a copula transform to a variant of kernel CCA using random projections, resulting in a O(n log n) complexity. Results on synthetic and real benchmark data show promising results for feature selection.

The paper is overall clear and pleasant to read. The good experimental results and simplicity of implementation suggest that the proposed method may be useful in complement to other existing methods.

The originality is limited, since it mostly combines several "tricks" that have been used in the past, namely the copula transformation to make the measure invariant to monotonic transformations (see eg Conover and Iman, The American Statistician, 35(3):124-129, 1981 or more recently the reference [15] cited by the authors), and the random projection trick to define a low-rank approximation of a kernel matrix.

Although the work is technically correct, the following points would require clarification:

- the authors insist that RDC is much faster to implement than kCCA, claimed to be in O(n^3) and taking 166s for 1000 points in the experiments. I am surprised by this, since in the original kCCA paper of Bach and Jordan an implementation in O(n) using incomplete Cholevski decomposition is proposed, and the authors claim there that it takes 0.5 seconds on 1000 points (more than 10 years ago). In fact, the incomplete Cholevski decomposition of Bach is a very popular approach to run kernel methods on large numbers of samples, similar to the random projection trick used by the authors. A natural question is then: to perform kCCA with large n, is it really better to use the random projection trick compared to the incomplete Cholevski decomposition?

- as pointed out by the authors, when the dimension k of random projections gets large the method converges to unregularized kCCA, so it is important that k is not too large because "some" regularization of kCCA is needed. Hence k seems to play a crucial regularization role, akin to the regularization parameter in kCCA. Is there any theoretical or empirical argument in favor of a regularization by k, compared to the classical kCCA regularization? An argument against using k is that, when k is not too large, the random fluctuations may lead to significant fluctuations in the final score, which is not a good property. In fact, although RDC fulfills all conditions in Table 1, one that is not fulfilled is that it is a stochastic number (ie, compute it twice and you get different values).
Summary: An interesting work combining several known ideas, but the comparison with the nearest cousin kCCA is not really fair and well-studied.

Submitted by Assigned_Reviewer_5

The RDC is a non-linear dependency estimator that satisfies Renyi's criteria and exploits the very recent FastFood speedup trick (ICML13). This is a straightforward recipe: 1) copularize the data, effectively preserving the dependency structure while ignoring the marginals, 2) sample k non-linear features of each datum (inspired from Bochner's theorem) and 3) solve the regular CCA eigenvalue problem on the resulting paired datasets. Ultimately, RDC feels like a copularised variation of kCCA (misleading as this may sound). Its efficiency is illustrated successfully on a set of classical non-linear bivariate dependency scenarios and 12 real datasets via a forward feature selection procedure.

I found the paper very clear and easy to understand. Even though the idea simply puts together the right pieces, it remains a significant contribution to the literature.

Some notes:
- I do not like that k was simply 'set to 10'. It seems to play a more important role than the paper implies.
- line 74: what do you mean by "same structure as HGR"?
- eq.(5): m -> n
- line 217: which is independent -> by independence
Summary: RDC is a straightforward and computationally efficient estimator of the HGR coefficient but the choice of k deserves more discussion.

I've read the author's rebuttal.

Submitted by Assigned_Reviewer_6

The authors propose a non-linear measure of dependence between two random variables. This turns out to be the canonical correlation between random, nonlinear projections of the variables after a copula transformation which renders the marginals of the r.vs invariant to linear transformations.

This work details a simple but seemingly powerful procedure for quantifying non-linear dependancies between random variables. Part of the key idea of the work is the use of non-linear random projections via the "random fourier features" of Rahimi and Recht (2007). The authors compare their new randomised dependence coefficient (RDC) against a whole host of other coefficients for measuring both linear and non-linear dependence. They also consider the suitability of the RDC for screening for variable selection in linear prediction problems. I think these are useful comparisons and experiments and they allow the reader to get a decent feel for the behaviour of RDC.

The overall exposition of the paper is clear and each component part of the procedure is clearly described. In this respect the schematic diagram in figure 1 is particularly useful. Similarly, the comparison between other dependence coefficients combined with the empirical results is very illuminating and suggests that the performance of RDC is very promising.

Although the authors choose to use RFFs, recently comparisons have been made with other schemes for generating random features have been made. Yang et al (2012) "Nystrom Method vs Random Fourier Features: A Theoretical and Empirical Comparison" flesh out some more of the theoretical properties of the difference between RFF and Nystrom features. One difference in particular is the difference in sampling scheme: Nystrom samples randomly in such a way that it takes the probability distribution of the data into account. In this sense it achieves a better estimate of the underlying kernel. If this is an important property of the proposed RDC, perhaps looking at Nystrom features would be interesting. It also would be interesting to see the effect of different types of non-linear projections where RFFs are limited to shift invariant kernels (with some other extensions), Nystrom features can be computed for any type of kernel function.

Also, Yang et al show empirically that there is a large improvement in predictive performance when more random features are used - it would be interesting to see what happens when k is increased.

I have some concerns about the theoretical treatment presented.
The discussion of the bound in (10) seems to be skipping over the tradeoff involving k. It seems that in order to drive the LC/sqrt(k) term to be small (which could be quite slow?), ||m||_F could grow very large compared with n but this is hidden by referring to the dependancy as O(1/sqrt(n)). As it stands I'm not sure the analysis is sufficient to complement the good empirical performance.

With regards to the properties of the RFF approximation for linear prediction, one possibility is that the approximation acts as a regulariser. For this reason, I am not sure that the type of generalisation bound used for prediction algorithms are completely appropriate to quantify the performance and analyse the behaviour of a dependence measure.

One small point: the reference to Recht is incorrectly referred to as "Brecht" in the text.
===============

I have read the author rebuttal and I am satisfied with the response. The authors should be sure to clarify the constraints in the CCA optimisation problem.
Summary: I think overall the work is extremely interesting and appears to work well empirically. As it stands I think the theoretical analysis is incomplete although that does detract too much from the impact and importance of the work.

Submitted by Assigned_Reviewer_7

This paper gives a new approach to nonlinear correlation. The approach consists of three steps: (1) copula transform, (2) random nonlinear projection, and (3) CCA. In each step, the authors give a theoretical guarantee to make the approach convincing. The authors demonstrate the usefulness of the approach using synthetic and real data set.

This is a well-written paper having high novelty. As the authors describe, the RDC score is easy to compute. Their experiments, Fig.4 in particular, clearly demonstrate the utility of the approach. I enjoyed reading the paper very much. Although calculating nonlinear correlation itself has not been the central topic in the machine learning community, I found it an interesting building block for more realistic tasks such as regression as one of the experiments already suggests. I recommend accepting the paper.
Summary: New and interesting approach to the metric of nonlinear correlation. The novelty is high.
Author Feedback

Author rebuttal: Dear reviewers,

Thank you for the supportive feedback and useful comments; we are delighted to see this positive response to our submission.

"comparison to kCCA with Cholesky decomposition"

This is a fair point. We downloaded the implementation of kCCA from Bach and Jordan. We used the default parameters (a moderate amount of them exist to fine tune the procedure) and the provided MEX/C compiled routines. The running times were much slower than RDC: 0.40, 3.24 and 43.80 seconds for respective sample sizes of 1000, 10000 and 100000. Preliminary power measurements indicate worse accuracy and higher sensibility to regularization than RDC. The implementation is also notably more involved (around 1000 lines of MATLAB). A detailed comparison will be added to the paper.

Conceptually, we see several advantages for random features:
1) For n samples of d dimensions and approximations of rank k, Fastfood expansions require less RAM (O(k) << O(kd)) and less CPU time (O(nk log d) << O(nk^2d)) than low-rank kernel matrix approximations such as Nystrom or Incomplete Cholesky Decomposition. This difference is crucial, for instance, when dealing with collections of images (large d) or big k.
2) Random-Feature expansions can outperform low-rank kernel matrix approximations (Le et al., 2013). This is because one might not want to focus on approximating the kernel in the first place (as in Nystrom), but to construct a good span of random basis functions for regression (as in random features).
This discussion will also be included in the paper.

"selection of k, number of random features"

This is the most interesting and challenging problem, which translates to the classical dilemma of (unsupervisedly!) choosing a regularizer or approximation rank: the more random features we allow, the more flexible the function class becomes. However, we experienced a rather satisfactory robustness against the number of random features: when iterating k from 5 to 500, we usually observed variations in the RDC score of less than one percent. However, this is not the case for the choice of the random basis function family (i.e., the kernel function in the kernel machine analogy), which seems of much greater importance. Thanks for pointing this out, a better illustration of this matter will be included in the paper.

"comparison to Nystrom"

Many thanks to reviewer for pointing out (Yang et al., 2012). We will try out the Nystrom idea.

"the norm of m in the theoretical analysis"

Given the constraints of CCA optimization, the L2-norms of alpha and beta are both constrained to 1, which effectively controls the Frobenius norm of m. The influence of k is incorporated to the bound analogously as in Rahimi and Recht (2008). Please note that the bound measures the distance between RDC and HGR when the latter is constrained to the function class F. This will be clarified in the paper.

"the proposed method is a concatenation of previously known techniques"

It is true that each of the steps that form RDC is previously known. But we don't think that, had someone used them in this way for dependency analysis in a paper on another matter, it would have been considered an "obvious" thing to do with no need for further analysis. Our paper motivates each individual step, analyses it theoretically, offers empirical evaluation and demonstrates the ease of implementation.

"random fluctuations"

As shown in Section 5, random fluctuations had little impact on RDC performance. Random variation is a valid theoretical concern, but also applies to other approximations needed in the large-scale setting, such as low-rank approximations to kernel matrices. This motivates the interesting idea of building distributions over RDC scores on a given dataset to yield a more powerful analysis of its value and confidence (as in bootstrap).